# Pannexin 1 Transgenic Mice: Human Diseases and Sleep-Wake Function Revision

**DOI:** 10.3390/ijms22105269

**Published:** 2021-05-17

**Authors:** Nariman Battulin, Vladimir M. Kovalzon, Alexey Korablev, Irina Serova, Oxana O. Kiryukhina, Marta G. Pechkova, Kirill A. Bogotskoy, Olga S. Tarasova, Yuri Panchin

**Affiliations:** 1Laboratory of Developmental Genetics, Institute of Cytology and Genetics SB RAS, 630090 Novosibirsk, Russia; korablevalexeyn@gmail.com (A.K.); irina_serova2004@mail.ru (I.S.); 2Laboratory of Mammal Behavior and Behavioral Ecology, Severtsov Institute Ecology and Evolution, Russian Academy of Sciences, 119071 Moscow, Russia; kovalzon@sevin.ru; 3Laboratory for the Study of Information Processes at the Cellular and Molecular Levels, Institute for Information Transmission Problems, Russian Academy of Sciences, 119333 Moscow, Russia; kcyu@yandex.ru (O.O.K.); ypanchin@yahoo.com (Y.P.); 4Department of Human and Animal Physiology, Faculty of Biology, M.V. Lomonosov Moscow State University, 119234 Moscow, Russia; marta.peckovva@gmail.com (M.G.P.); k.bogotskoy@yandex.com (K.A.B.); ost.msu@gmail.com (O.S.T.); 5Department of Mathematical Methods in Biology, Belozersky Institute, M.V. Lomonosov Moscow State University, 119234 Moscow, Russia

**Keywords:** pannexin 1, CRISPR/Cas9 method, sleep-wake cycle

## Abstract

In humans and other vertebrates pannexin protein family was discovered by homology to invertebrate gap junction proteins. Several biological functions were attributed to three vertebrate pannexins members. Six clinically significant independent variants of the *PANX1* gene lead to human infertility and oocyte development defects, and the Arg217His variant was associated with pronounced symptoms of primary ovarian failure, severe intellectual disability, sensorineural hearing loss, and kyphosis. At the same time, only mild phenotypes were observed in Panx1 knockout mice. In addition, a passenger mutation was identified in a popular line of Panx1 knockout mice, questioning even those effects. Using CRISPR/Cas9, we created a new line of Panx1 knockout mice and a new line of mice with the clinically significant Panx1 substitution (Arg217His). In both cases, we observed no significant changes in mouse size, weight, or fertility. In addition, we attempted to reproduce a previous study on sleep/wake and locomotor activity functions in Panx1 knockout mice and found that previously reported effects were probably not caused by the Panx1 knockout itself. We consider that the pathological role of Arg217His substitution in Panx1, and some Panx1 functions in general calls for a re-evaluation.

## 1. Introduction

The pannexin protein family was discovered in 2000 as a result of a comparison of human and invertebrate genes [1]. In subsequent years, three vertebrate pannexin genes were cloned and characterized [2,3,4,5,6]. The members of this family, especially pannexin 1, are implicated in a number of vital biological functions and the development of several pathological mechanisms [7,8,9,10]. It was previously recognized that pannexin channels, such as connexin hemichannels, consist of six subunits. Four new studies using cryo-electron microscopy [11,12,13,14] have shown that pannexin channels actually consist of seven subunits. Elucidating the structure of pannexin 1 with high resolution allowed a more accurate analysis of the possible roles of known mutations in the functioning of this protein.

Pannexins play a key role in intercellular communications because they are (1) capable of forming intercellular gap junctions (GJ), (2) functioning in the plasma membrane as channels highly permeable for adenosine triphosphate (ATP) and other important signaling molecules, (3) function as calcium leak channels in the endoplasmic reticulum membrane [4]. Connexins are an established family of GJ proteins that are specific to vertebrates and invertebrate pannexins (known as innexins) also form GJ’s [6]. The formation of intercellular GJ’s by mammalian pannexins was confirmed in several publications [2,4,15,16], although the existence of pannexin GJ’s in mammals in vivo is disputed [6]. Glycosylation in the extracellular portion of the Panx1 protein is considered to be a barrier for GJ formation [17]. The general question of the number of GJ protein families is complicated by the observation that some Metazoans can form GJ’s while lacking both connexins and pannexins. These results suggest the existence of a third hypothetical family of gap junction proteins. These proteins can actually co-exist with connexins and pannexins in mammals, for example, in humans [18].

The functioning of pannexins as channels highly permeable for purines in the plasma membrane is well documented [19] and the current consensus in the field identifies pannexins as the major pathway of ATP release into the extracellular media. It was suggested that this property of Panx1 is responsible for its role in the sleep-wakefulness cycle [20,21] and in the regulation of vascular tone [22,23,24]. It is also becoming evident that pannexins play an important role in a wide range of medically significant processes, such as apoptosis, inflammation, ischemia, and tumor genesis [8,25,26], as well as neuropathic pain [27].

On the other hand, genomic studies linking *PANX1* to human pathologies have come into sight. According to new data (https://www.omim.org/entry/608420, accessed date 15 May 2021), five variants of *PANX1* are clinically significant: Gln392Ter, Cys347Ser, Lys346Glu, and 9-BP DEL, NT61 lead to human oocyte development defects and infertility [28,29]. Arg217His variant was reported in the case with pronounced symptoms of primary ovarian failure, severe intellectual disability, sensorineural hearing loss, and kyphosis [30]. At the same time, only mild phenotypes were observed in Pannexin1 knockout mice [20]. In addition, it was reported that a passenger mutation in the Casp11 gene was identified in a popular line of pannexin 1 knockout mice, questioning even those effects [31]. Here we reduplicated pannexin 1 knockout mice using the CRISPR/Cas9 method and reproduced substitution at position 217 (p.Arg217His) in the mice model. In both cases, there were no obvious changes in phenotypes such as size, weight, or fertility. In addition, we compared the sleep/wake and locomotor functions in the new strain of pannexin 1 knockout mice with the strain with the (Arg217His) mutation and their genetic background (C57Bl/6J) and found no differences between the three strains. In this regard, we believe that the pathological role of *PANX1*, the Arg217His substitution, and some functions of *PANX1*, in general, require re-evaluation. Although there are known cases of mutations whose phenotypes differ radically in humans and mice (for example, the connexin 26 mutation [32,33].), our data suggest that human multisystem dysfunction [30] is not associated with the Arg217His replacement, as suggested previously.

## 2. Results

### 2.1. Generation of Mouse Models

#### 2.1.1. Generation of the *Panx1* Knockout Mice Strain

To knock out the *Panx1* gene, we chose the strategy of deleting its exons 3 and 4. Dvoriantchikova et al. previously showed that the deletion of these exons results in protein disappearance [34]. According to the previously described scheme [35], we used the CRISPR/Cas9 system to carry out these genetic modifications. To increase the efficiency of obtaining deletions, we also used the short single-stranded DNA template (ssODN). Several nucleotides that form a restriction site for the HindIII restriction enzyme have been added to the ssODN sequence for genotyping convenience. The design of genetic modifications and genotyping strategy are shown in Figure 1A.

To obtain mice with the *Panx1* gene knockout cytoplasmic microinjections of 317 zygotes of the inbred C57BL/6J line were performed with a solution containing 25 ng/μL gRNA-1, 25 ng/μL gRNA-2, 50 ng/μL mRNA-spCas9, and 100 ng/μL ssODN-1. One hundred 58 injected zygotes were transplanted into pseudopregnant CD-1 females. As a result, 34 mice were born (21.52% of the transplanted embryos), 29 survived. Among all born mice, we identified 28 animals with deletions of exons 3 and 4. Moreover, seven mice were homozygous for the deletion, and the remaining 21 were heterozygotes. Since the ssODN template also had a restriction site for HindIII, we could discriminate between deletion events by two different mechanisms, homology-directed repair (HDR) and non-homologous end-joining (NHEJ) (Figure 1B,C). If we assume that in 34 animals, 68 alleles could undergo modification, we can conclude that in more than half of the cases, deletions occur, and the proportion of the HDR mechanism is about 13% (Figure 1D). This result is in suitable agreement with the high activity of homologous recombination in the mouse zygote [36].

#### 2.1.2. Generation of Arg217His Mice

Human and murine Panx1 proteins have 86% identity. The amino acid arginine 217 for which the Arg217His mutation has been described is located in a region of 7 conserved amino acids in mice and humans (Figure 2A). Therefore, we decided to reproduce this mutation in the mouse by replacing the single amino acid Arg216His in the mouse protein. An alternative strategy would be to replace all or part of the coding sequence of the mouse gene with a human one with an Arg217His mutation. However, this approach is much more challenging to implement. Moreover, such humanization can potentially introduce additional side effects because the amino acid sequence of human and mouse proteins differs, and these differences will affect interactions with other proteins in mouse cells.

To generate Arg217His mice using CRISPR/Cas9, we introduced a single break in the gene region corresponding to codon 216, and the ssODN was added, which differed from the wild type by two nucleotides, changing the CCG (R) codon to the GTG (H) codon. Thanks to the experiment’s design for genotyping with restriction endonucleases, we evaluated the efficiency of genomic editing, namely, to detect unmodified wild-type alleles, alleles resulting from HDR using the ssODN template, as well as alleles containing indels resulting from NHEJ. The design of the genetic modifications and the analysis of subsequent genomic changes are shown in Figure 2B,C.

To obtain mice with the Arg217His substitution in the Panx1 gene cytoplasmic microinjections of 320 zygotes of the C57BL/6J inbred line were performed with a solution containing 25 ng/μL gRNA-3, 50 ng/μL mRNA-spCas9, and 100 ng/μL ssODN-2. One hundred 84 injected zygotes were transplanted into pseudopregnant CD-1 females. As a result, 47 mice were born (25.54% of the transplanted embryos), 44 survived. Among all 47 mice, using PCR analysis and subsequent restriction analysis, 16 animals with the target modification R216H were identified, among which two mice were homozygous. The remaining 31 mice did not have the target modification. However, 28 mice were homozygous for indels, and 3 were heterozygotes. Moreover, we did not find a single animal that had both wild-type alleles (Figure 2D). From the entire experiment on obtaining mice with the R216H modification, 47 mice were born, which means there were 94 alleles that the CRISPR system could modify. So, it is possible to calculate the efficiency of the CRISPR system and the proportion of two different ways of double-strand breaks repair—HDR and NHEJ. As a result, the CRISPR system’s efficiency is about 95%, and the probability of double-strand break repair by the HDR mechanism is about 19% (Figure 2E).

After generating the F0 generation of mice, we selectively performed Sanger sequencing to refine the DNA sequences in the region of target modifications. After that, F0 mice were selected to establish two new homozygous mouse strains:C57BL/6J-Panx1^em1Koral/Icg^—a strain of mice with knockout of the *Panx1* gene by deletion of 3 and 4 exons;C57BL/6J-Panx1^em2Koral/Icg^—a line of mice with a target change in the fourth exon of the *Panx1* gene: at position # 216, the CGG codon (codes for arginine) was replaced by the CAC codon (codes for histidine).

Further in the text, these two lines are referred to as “KO2020” and “ST2020”, respectively, and their genetic background C57BL/6J strain is referred to as “WT2020”.

After establishing homozygous lines, we performed whole-genome sequencing for one animal from each strain. Data analysis showed that the target mutation is indeed in a homozygous state (Appendix A). It is known that CRISPR/Cas9 can have an off-target activity that can lead to unwanted mutations [37]. Therefore, in each strain, for each gRNA, we visually inspected the top 50 predicted off-target sites of the genome. We did not find indels or point mutations in any of them.

### 2.2. Gross Characteristics of Two Generated Strains of Mice

Two generated mouse strains were bred during a year to study their fertility and growth rate. No differences in litter size were observed among KO2020, ST2020, and WT2020 strains. The percentage of male pups per litter did not differ as well (Table 1).

On external examination of sexually mature (8-week-old) progeny, there were no differences between the groups (Figure 3). Along with that, male body weight in both transgenic strains was somewhat smaller compared to the wild-type group (Table 1). In addition, males from the substitution group were smaller than knockout males. The eight-week-old female weight did not differ between wild type and knockout groups but was slightly less in the substitution group compared to the wild-type group. Then one or two males were randomly selected from the obtained litters and grown to 20 weeks of age. Intergroup differences in body weight persisted at an older age: body weight was decreased in the row wild type—knockout—substitution (Table 1).

### 2.3. The Results of Polysomnographical Study

Figure 4 shows the results of visual scoring of 24-h polysomnograms in WT2020 (wild type) and two mouse strains generated using the CRISPR/Cas9 method: KO2020 (Panx1 knockout) and ST2020 (Panx1 substitution). For comparison, we added the data from our previous study [20] performed on the Panx1^−/−^ strain developed by Dvoryantchikova et al. [34] (KO2017) and their control group (WT2017, C57Bl/6J mice from the vivarium of the Severtsov Institute). In addition, the “conventional” data from wild-type mice published in [38] are shown (group WT2000). As can be seen, no significant difference was found among the groups except for the group KO2017, in which wake duration was significantly increased, and slow-wave sleep duration was significantly reduced compared to their durations in the corresponding control group (WT2017). The latter effect reported in our previous communication [20] is probably due to a passenger mutation in the Casp11 gene (see Discussion).

The hourly distribution of wakefulness (Figure 5A) and the 12-h summary (Figure 5B) demonstrate its normal increase in the dark period and decrease in the light period of the 12L/12D cycle in three studied groups. Although the wake percentage in the ST2020 group shows the shift in the same direction as the KO2017 group, that is, an increase in the dark and decrease in the light period of nychthemeron, in the former group, it was small and not statistically significant. In general, the lack of any statistically significant differences among the groups is clearly seen.

Importantly, locomotor activity did not differ among the three studied groups of mice (Figure 6). This observation suggests that neither the Panx1 knockout nor the Arg217His replacement in Panx1 affected the musculoskeletal system of the mice.

## 3. Discussion

This work is not the first presentation of the genetic modification of the *PANX1* gene [34,39,40,41,42]. A comprehensive list of Panx1 knockout strains, including publication references and reported phenotypes, is presented on two Internet sites (https://www.alliancegenome.org/gene/MGI:1860055#summary, accessed date 15 May 2021 and http://www.informatics.jax.org/marker/phenotypes/MGI:1860055, accessed date 15 May 2021). In our work here, we were partially motivated by a clinical case of human multisystem dysfunction [30] potentially associated with a homozygous Panx1 mutation. Shao et al. [30] also reported that Arg217His replacement in Panx1 results in the protein loss of function. A new study reported that in HEK-293 cells transfected with Panx1 Arg217His mutation construct the Panx1 channel current was reduced compared to the wild-type construct but no difference was found if Arg-217-His mutant and the wild-type Panx1 were C-terminal tail truncated [43].

It was reasonable to suggest that the ARG to HIS missense substitution would reproduce the effects of Panx1 knockout that, by definition, results in loss of function of Panx1. Yet, no such obvious and severe phenotypes were reported in previously constructed Panx1 KO mice [34,39] or in our study. It is important to note that the proband of the human clinical case [30] was most probably infertile, but in the murine model, homozygous Panx1 KO animals normally reproduce, as do double homozygous Panx1^−/−^/Panx2^−/−^ KO animals [44]).

Presumably, the conclusion regarding the loss of function effect of Arg217His mutation is incorrect. What if it results in a gain of function? Six independent cases of presumably gain-of-function Panx1 mutations were recently reported [28,29]. These variants lead to human oocyte development defects and infertility, but no multisystem dysfunction was reported in carriers.

One more explanation of this discrepancy is that Arg217His mutation is not functionally equivalent to gene loss in some other way, not corresponding to simple loss or gain of function. So, we generated the mouse strain with ARG substituted to HIS in position, homologous to position 217 that was mutated in human Panx1. These mice also normally reproduced and displayed no multisystem dysfunction as skeletal defects, including kyphoscoliosis, reported in the human case study [30]. Alternatively, the Panx1 mutation phenotypes may differ radically in humans and mice. For example, connexin 26 KO is lethal in mice, whereas a seemingly equivalent homozygous gene loss in humans (35delG, causes a frameshift with premature termination of the protein at the twelfth amino acid) is not lethal and results in non-syndromic hearing loss (DFNB1) [32,33].

Based on new structural data [11,12,13,14] we can more accurately localize the position of Arg217 to the middle of the S3 transmembrane domain that is not directly contacting the conductance pore. Although arginine is more positively charged than histidine, there is no obvious evidence why this substitution will result in a dramatic change of the Panx1 function. Interestingly this position is conserved in mammalian Panx1 proteins but is different in Panx2 and Panx3 paralogs, and in Panx2, it corresponds to His as in the human mutation. Considering this possibility, our data suggest that human multisystem dysfunction [30] may not be associated with Arg217His replacement but may have a different etiology. Of note, the role of other genomic variants in multiorgan syndrome identified in proband whole-exome sequencing was refuted by the authors’ analysis partially because they were inherited from her asymptomatic mother or father [30].

On the other hand, some Panx1 mutations that lead to human oocyte development defects and infertility fit well with structural data. For instance, the Δ21–23 disease mutation [29] will affect the narrow part of the tunnel resulting in a gain-of-function phenotype. This was experimentally confirmed by a Δ21–27 deletion [14]. All clinically significant genomic variants discussed here were found in individual clinical cases and are extremely rare or not found in independent massive genomic screens. At the same time, there are 255 Missense and 11 pLoF SNVs observed in the gnomad database (https://gnomad.broadinstitute.org/gene/ENSG00000110218?dataset=gnomad_r3, accessed date 15 May 2021).

Early stop gained or frameshift mutants may result in protein inactivation. They were found on single occasions and were heterozygous. So, although such mutants are present in the human population, they are very rare and probably have no phenotypes as compensated by the second healthy allele. Stop gained or frameshift mutants in the C-terminal may result in a gain-of-function. It was shown that the cleavage of the C-terminal tail (CTT) either during apoptosis or experimentally by caspase 3 or 7 results in channel activation and ATP release [45,46]. According to the gnomad database, several heterozygous SNVs of this type were detected in CTT (Val297HisfsTer31 frameshift, Pro298HisfsTer31 frameshift, Arg300Ter stop gained) and can result in C-truncated protein or distort its pore-blocking function.

Before the invention of efficient zygote microinjection, CRISPR-Cas9 based protocols that targeted mutagenesis in mice relied strongly upon germline transmission competent embryonic stem cells (ESC) lines derived from the 129 strain of mouse [47]. Although numerous backcrossing strategy is used to omit ESC-derived genetic background effects of this approach, genetically engineered animals can retain passenger mutations in the region closely flanking the targeted gene. Comparative genomics analysis between C57BL/6 and the 129 strains [31] identified multiple passenger mutations (1084 genes are predicted to be aberrantly expressed). In the popular Panx1 knockout strain [34] Casp11, an inactivating passenger mutation was found that can produce “false-positive” phenotypes. Murine caspase-11, and its human homologs caspase-4 and caspase-5, are believed to be an inflammatory caspase, along with caspase 1, with a role in the immune system.

The role of pannexin 1 in inflammation and its possible interference with Casp11 were not studied in this paper and are discussed elsewhere [48]. Yet one aspect of inflammation and the Casp11 inactivating passenger mutation can be relevant to the discrepancy in results of the sleep-wake study reported here and previous work [20]. Several elements of inflammation machinery can affect the sleep-wake cycle.

IL-1β applied centrally or to the periphery enhances NREM sleep in rabbits, rats, and mice [49,50,51]. Activation of inflammasomes facilitates the release of interleukin-1β (IL1β) and IL18. The NLRP3 inflammasome was reported to modulate sleep [52]. We hypothesize that some alterations of the sleep-wake cycle in Pannexin1 knockout with the Casp11 inactivating passenger mutation are related to modification of the Casp11 gene. As Pannexin1 is likely involved in the inflammation response, it will be interesting to study sleep-wake effects in a “clear” Casp11 knockout model.

## 4. Materials and Methods

### 4.1. CRISPR/Cas9 Design and Cytoplasmic Microinjection

gRNA was designed using the Benchling online service (https://benchling.com/, accessed date 15 May 2021). To obtain mice with the replacement of R216H (replacement of the 216th codon arginine with histidine), one gRNA-1 was selected in the Panx1 gene, and two gRNAs, gRNA-2 and gRNA-3, were selected for the knockout of the Panx1 gene (Table 2). Using the online service “Primer Blast” (https://www.ncbi.nlm.nih.gov/tools/primer-blast/index.cgi, accessed date 15 May 2021), primers were selected for genotyping target mutations in the Panx1 gene (Table 2).

To obtain mice with the target R216H mutation in the Panx1 gene and to increase the proportion of knockout mice for the Panx1 gene, short single-stranded DNA was used (Table 2) [35].

For the synthesis of gRNA, a DNA template containing the T7 promoter was prepared by PCR. Then, after purification, the DNA template was used for in vitro gRNA transcription using the HiScribe T7 High Yield RNA Synthesis Kit (NEB, E2040S, Ipswitch, MA, USA). The gRNA was purified on RNA Clean and Concentrator-25 columns (Zymo Research, R1017, Irvine, CA, USA) [53]. SpCas9 messenger RNA was purchased from a commercial manufacturer (GeneArt CRISPR Nuclease mRNA, Thermo Fisher Scientific, Shanghai, China).

Cytoplasmic microinjection of zygotes was performed using standard techniques that are widely used in transgenesis [35,53].

### 4.2. Recording of the EEG and Motor Activity in Chronically Instrumented Mice

All animal procedures were approved by the Ethics Committee of Severtsov Institute Ecology and Evolution, Russian Academy of Sciences (protocol #3, approval date 19 June 2017). Chronic electrodes were implanted to WT2020, KO2020 and ST2020 mice (eight mice per group) aged 2.5–3 months and weighing 25–30 g under general anesthesia (Zoletil, Virbac, France, 35 mg/kg, i.m.) for registration of the EEG of the frontal and parietal areas of the neocortex. After the surgery, the animals were placed in individual soundproof boxes, in which they were kept at the temperature of 22–24 °C and 12 h/12 h illumination mode (09:00–21:00—bright white light, 21:00–09:00—dim red). Water and standard rodent food were available ad libitum. After a week of post-surgery recovery and adaptation to the experimental environment, a 24-h recording of EEG and locomotor activity was performed.

Each animal was permanently connected through a flexible cable to the input of a 2-channel miniature autonomous digital telemetric a biopotential amplifier (30 × 25 × 4 mm in size and 5 g in weight, designed by A.A.Troshchenko (http://biorecorder.com/ru/br8v1.html, accessed date 15 May 2021), equipped with a 3D accelerometer. The amplifier board was connected by an elastic connection to the power battery and, together with it, suspended from the rod above the camera by means of a miniature rotating karabiner. This design allowed us to record EEG without restricting the freedom of animal movements since a biopotential amplifier board with a built-in accelerometer freely oscillated in three dimensions and, therefore, responded even to small movements of a mouse. EEG was recorded with a sampling rate of 250 Hz and motor activity of −50 Hz. The amplifier signals were transmitted to a recording computer via the Bluetooth channel. Visual scoring of the obtained polysomnograms (2 EEG channels and three accelerometer channels) was carried out offline for 20-s epochs using a special program created on the basis of an open-source EDF browser [54].

According to generally accepted criteria for rodents [38], the states of wakefulness, slow-wave sleep, and REM sleep were distinguished. Comparison of the hourly percentage of states of wakefulness, slow-wave sleep, and REM sleep in three groups of mice studied so far was carried out; the results were compared with the literature data. Statistical analysis of the results was carried out using Kruskal–Wallis nonparametric ANOVA and post hoc Dwass-Steel-Critchlow-Fligner pairwise comparisons.

## 5. Conclusions

The data presented in this communication provide the basis for the following important conclusions.

(1)The previously observed changes in the wakefulness cycle and the behavior of Panx1 knockout mice [20] were apparently associated not with the target, but with passenger mutation(s). Hence, our hypothesis on the possible role of the pannexin 1 protein in the regulation of the sleep-wakefulness cycle [21] should be revised.(2)The absence of noticeable changes in such fundamental characteristics, as an appearance, general behavior, reproductive function, and sleep-wakefulness cycle in mice with an artificial point mutation of the *Panx1* gene (Arg217His) doubts the previously reported data on the association of this mutation with multisystem dysfunction in the human organism [30]. Consequently, our hypothesis about the presence of a certain number of such patients [55] is questioned.(3)Further studies of the role of pannexins at the systemic level are needed.

## Figures and Tables

**Figure 1 ijms-22-05269-f001:**
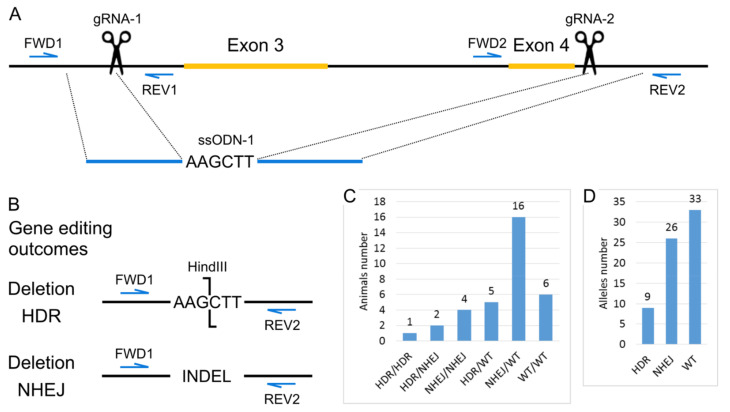
Generation of the Panx1 knockout mice strain. (**A**) Scheme of the modified region of *Panx1* gene. Positions of Cas9 generated double-stranded breaks are indicated as scissors. (**B**) Gene editing outcomes. The scheme for PCR and restriction enzyme-based genotyping strategy. (**C**) Genotypes of mice from the cytoplasmic microinjections experiment. (**D**) Estimation of the number of alleles repaired by particular DNA break repair mechanisms.

**Figure 2 ijms-22-05269-f002:**
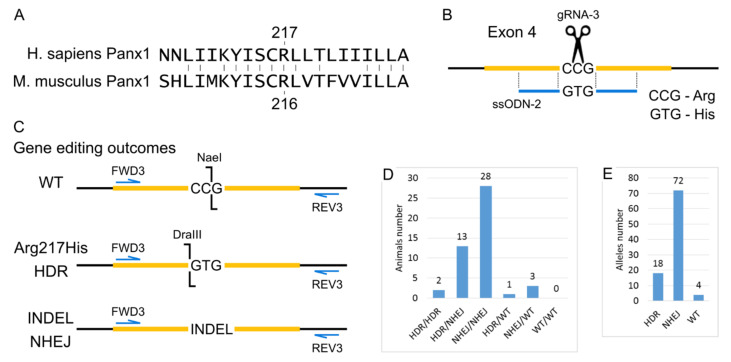
Generation of Arg217His Mice. (**A**) Local alignment of the human and mouse Panx1 proteins in the region around Arginine (R) 217. (**B**) Scheme of the modified region of *Panx1* gene PoScheme 9 generated double-stranded break is indicated as scissors. (**C**) Gene editing outcomes. Scheme for PCR and restriction enzyme-based genotyping strategy. (**D**) Genotype of mice from cytoplasmic microinjections experiment. (**E**) Estimation of alleles number repaired by particular DNA break repair mechanisms.

**Figure 3 ijms-22-05269-f003:**
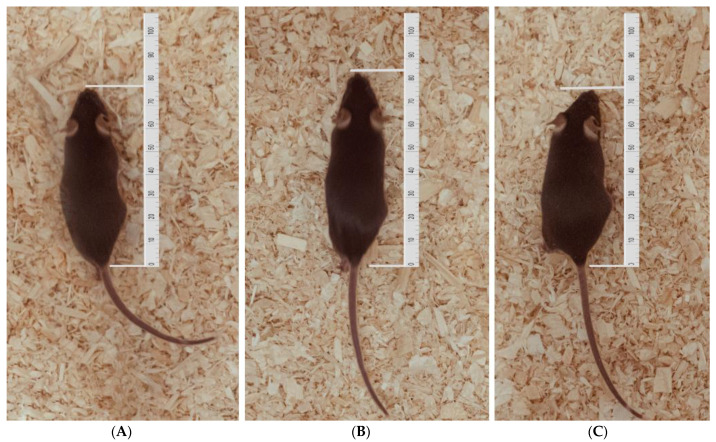
Male mice from three strains at the age of 8 weeks. (**A**) WT2020 (wild type); (**B**) KO2020 (Panx1 knockout); (**C**) ST2020 (Panx1 substitution).

**Figure 4 ijms-22-05269-f004:**
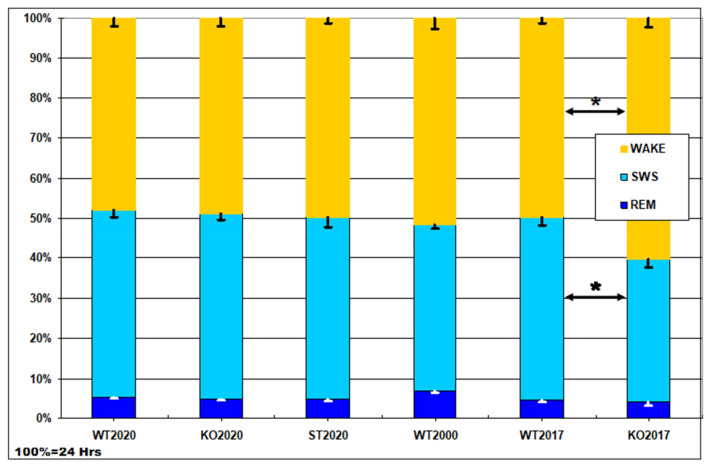
Daily percentage of wake, slow-wave sleep (SWS), and rapid eye movement (REM) sleep in six groups of mice: WT2020 (wild type, N = 8); KO2020 (Panx1 knockout, N = 8); ST2020 (Panx1 substitution, N = 8); WT2000 (wild type group from the study by Huber et al. [38], N = 11); WT2017, C57Bl/6J mice from vivarium of the Severtsov Institute (N = 10); KO2017, the Panx1^−/−^ strain developed by Dvoryantchikova et al. [34] (N = 13). Y axis – percentage of the daytime (100% = 24 h, Hrs). The data are given as mean ± S.E.M.; * *p* < 0.05 KO2017 vs. WT2017, Mann–Whitney U-test.

**Figure 5 ijms-22-05269-f005:**
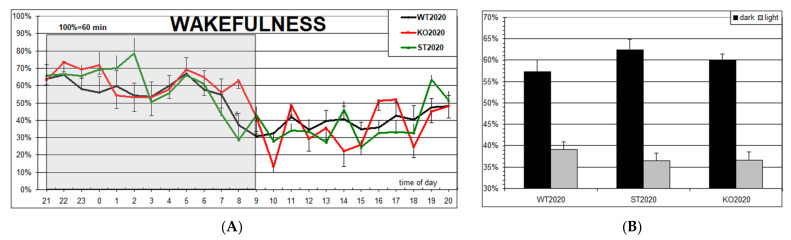
(**A**) Hourly percentage of wakefulness in three groups of mice (M ± S.E.M.): WT2020 (wild type, N = 8); KO2020 (Panx1 knockout, N = 8); ST2020 (Panx1 substitution, N = 8). The dark period in the chambers is shadowed. (**B**) 12-h summary of the data.

**Figure 6 ijms-22-05269-f006:**
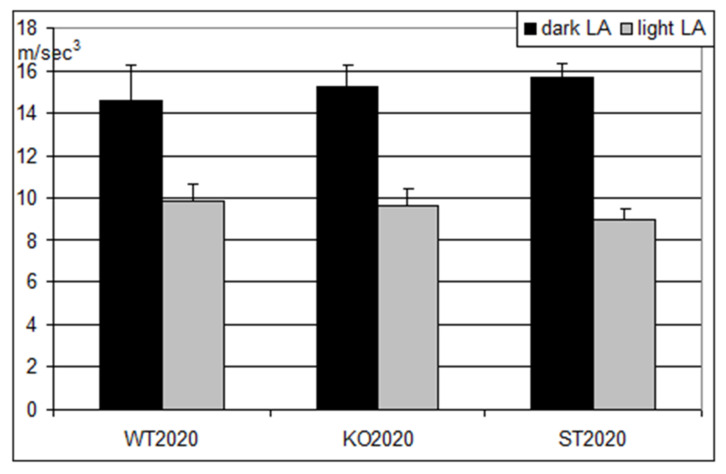
Locomotor activity in three groups of mice (M ± S.E.M.) during the dark (black columns) and the light (gray columns) periods of the day. WT2020 (wild type, N = 8); KO2020 (Panx1 knockout, N = 8); ST2020 (Panx1 substitution, N = 8). On the ordinate axis, mean hourly reading of accelerometer (m/s^3^).

**Table 1 ijms-22-05269-t001:** Characteristics of litters and body weight of grown progeny from these litters in three studied strains.

Parameters	WT2020	KO2020	ST2020
Number of pups per litter	6.3 ± 2.1	5.8 ± 2.5	5.8 ± 2.2
Percentage of males	54 ± 13	52 ± 23	42 ± 30
Male body weight at 8-week age, g	26.8 ± 1.9	25.1 ± 2.4 *	22.0 ± 2.5 *^,#^
Female body weight at 8-week age, g	19.8 ± 1.1	18.2 ± 1.5	17.5 ± 0.6 *
Male body weight at 20-week age, g	30.1 ± 2.7	27.5 ± 2.7 *	25.7 ± 2.0 *^,#^

Number of litters: WT2020—12; KO2020—24; ST2020—24; number of 20-week-old males (bottom line): WT2020—16; KO2020—16; ST2020—14. The data are given as mean ± S.D.; * *p* < 0.05 vs. WT2020, # *p* < 0.05 vs. KO2020 (one-way ANOVA with Bonferroni post hoc test).

**Table 2 ijms-22-05269-t002:** DNA oligos and Cas9 gRNA target sites used in this study.

Name	Sequence
gRNA-1	GGGACGCTGGCTGATCATGT (TGG)
gRNA-2	GCGAGTTCGTCCCTGAAATG (AGG)
gRNA-3	GAAATACATTAGCTGCCGGC (TGG)
ssODN-1	GACATGGCCTGGATCATGGACTTGCTACAGCTCAGCTACACAAGATGAAGAGAGCCAACAAGCTTCAGGGACGAACTCGCATCCCCACGTCACTGACAGACACCTGCCTGCTGCCTGTTGATT
ssODN-2	GAGGCTGAAGTAATAGCTCAAGTAGATACATGCCAACAGTATAACCACAAATGTCACCAGGTGGCAGCTAATGTATTTCATGATTAAATGACTAGAGTTCTTTTTTGTCTTCAAGTACTGCTC
FWD1	TAGCCCACTGTGAAGGGACT
REV1	CGGTTTCTAGCACCCCACAT
FWD2	ACACCCTACCTACCCCCTTC
REV2	CTGACATCCCTCTGGTCTGC
FWD3	ATGCCCAGGTTTGTCAGGAG
REV3	CAGTGTTGACAATGCCCGTG

## Data Availability

The next-generation sequencing data can be accessed online at Sequence Read Archive under the BioProject accession code PRJNA700445. All other data generated during this study are available from the corresponding author on reasonable request.

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
