# Peer review of "Pannexin 1 Transgenic Mice: Human Diseases and Sleep-Wake Function Revision"

_ijms, 2021, doi:10.3390/ijms22105269_

Round 1
Reviewer 1 Report
Battulin et al. studied Pannexin 1 transgenic mice and revised the links of PANX1 mutations with human pathologies. They used a CRISPR/Cas9 system to created Panx1 knockout mice and mice with the clinically significant Panx1 substitution (Arg217His).
In their models they could not reproduce major human pathologies linked to PANX1 gene deregulation such as infertility and sleep-awake functions, and suggested that Arg217His mutation may rather lead to gain-of-function, than funtional loss.
The manuscript is properly written, it is seem to be based on reliable models and the results support their conclusions.
However, it is known that KO animal models do not always reflect the severity of human pathologies. They also mention the case of Cx26. Another example were GJA1 KO animals that could be born despite Cx43 playing a massive role in coordinating action potentials in the myocardium, probably due to some compenstion by other Cx isotypes.
Is there any evidence for similar compensation by Panx2 or Panx3 in their model, which may not be activated in the reported human pathologies?
Are there any information what other passenges mutations were revealed in the published PANX1 human cases?
Though it was not significant, but the fluctuation ranges of wakefulness of PANX1 KO and PANX1 Arg217His substituted animals was obvious in Fig 5. compared to wild type animals.
Minor notes:
„Several other? functions were also attributed”… (no statement on any function before this sentence) Did you mean „severeal functions ?...
…”reported a passenger mutation „..The kind of mutation and it functional consequences must be a bit more specified even in the introduction to support the „questioning” of the published PANX1 mutation pathology.
Reviewer 2 Report
This manuscript describes “the making of” and the initial phenotyping of a new line of Panx1 knockout mice as well as a new line of mice with the clinically relevant Panx1 substitution. Although the data are essentially negative, the study is certainly innovative and informative for researchers in the Panx1 research field. However, to prevent further confusion in this research field, it is essential that the discussion of the manuscript includes in a broader comparison of the various Panx1 mouse models that are “presently on the market” than the Russian models only. In addition, the authors must use the recommended and commonly adopted terminology for pannexin channels.
Specific comments:
- Introduction, line 30: It is now commonly accepted in the pannexin research field that the terminology “pannexin hemichannels” should not be used anymore (see for example: doi: 10.4161/chan.5.3.15765). To prevent further contamination of the pannexin literature, please change this throughout the manuscript for “pannexin channels”. Alternatively, when used to indicate the structure of pannexin channels the term “pannexons” may also be used (the latter referring to the structural analogy with connexons).
- Introduction, line 35: Pannexins do not form gap junctions. Gap junctions are only formed by connexins. It is now widely recognized that the “pairing of pannexons” leading to gap junction-like currents in Xenopus oocytes was due to an artefact typical for this expression system only. Again, in order to prevent further contamination of the pannexin literature, please lease remove this statement from the manuscript.
- Page 3, lines 95-97: it is preferred to illustrate the species-specific differences in Panx1 amino acid sequences between mouse and human around this mutation, and better justify the choice for creating this specific mutation in the mouse in a figure.
- Figure 3 illustrates the measuring of he length of the three strains of mice at the age of 8 weeks. However, this data is not given in Table 1. Please add this data.
- Page 6, lines 177-178: The conclusion on normal circadian rhythmicity in these 3 strain is over-stated. For this conclusion, a more thorough investigation should be done. Please rephrase.
- The discussion on the passenger mutations associated to the Panx1 knockout strains is very important. The mice of the Shestopalov group is used as an example (doi:10.1371/journal.pone.0031991). It is a missed opportunity to limit this discussion point only “false positive phenotypes” only to the Panx1 knockout mice of Russian origin. The discussion must be broadened and should include an analyses of “the making of” all Panx1 knockout mouse models that are “presently on the market”. The easiest would be to make a an extra Table stating this information and includes a statement on the possibility for similar “false positive phenotypes” for each model. Such a Table would definitively further the Pannexin research field and would potentially result in many citations of this paper once it is published (which will be very much appreciated by IJMS)
- Conclusion 2, line 307-312: this statement in in conflict with an earlier statement in the discussion of the manuscript (lines 220-223) that the point mutation is in S3 transmembrane domain of the protein that is not directly contacting the conductance pore. Please revise.
- Conclusion 3, line 331: Please generalized this statement for its impact worldwide rather than in the Russian Federation only.
Typographical mistakes:
- Introduction, line 36: leek or leak ?
- Table 1: last column should be labeled ST2020 ?
- Page 7, line 185: duffer or differ ?
Round 2
Reviewer 2 Report
The authors adequately addressed my concerns. I have no further comments.
Author Response
We appreciate the time and effort that you have dedicated to providing your valuable feedback on the manuscript.